DOI: 10.1038/s41467-017-00373-2　　**OPEN**

# Unpredictability of escape trajectory explains predator evasion ability and microhabitat preference of desert rodents

Talia Y. Moore[1,2,3,4], Kimberly L. Cooper[5], Andrew A. Biewener[1,2] & Ramanarayan Vasudevan[6,7]

Mechanistically linking movement behaviors and ecology is key to understanding the adaptive evolution of locomotion. Predator evasion, a behavior that enhances fitness, may depend upon short bursts or complex patterns of locomotion. However, such movements are poorly characterized by existing biomechanical metrics. We present methods based on the entropy measure of randomness from Information Theory to quantitatively characterize the unpredictability of non-steady-state locomotion. We then apply the method by examining sympatric rodent species whose escape trajectories differ in dimensionality. Unlike the speed-regulated gait use of cursorial animals to enhance locomotor economy, bipedal jerboa (family Dipodidae) gait transitions likely enhance maneuverability. In field-based observations, jerboa trajectories are significantly less predictable than those of quadrupedal rodents, likely increasing predator evasion ability. Consistent with this hypothesis, jerboas exhibit lower anxiety in open fields than quadrupedal rodents, a behavior that varies inversely with predator evasion ability. Our unpredictability metric expands the scope of quantitative biomechanical studies to include non-steady-state locomotion in a variety of evolutionary and ecologically significant contexts.

[1] Department of Organismic and Evolutionary Biology, Harvard University, 26 Oxford Street, Cambridge, MA 02138, USA. [2] Harvard Concord Field Station, 100 Old Causeway Road, Bedford, MA 01730, USA. [3] Department of Ecology and Evolutionary Biology, University of Michigan, 1109 Geddes Ave, Ann Arbor, MI 48109, USA. [4] Museum of Zoology, University of Michigan, 1109 Geddes Ave, Ann Arbor, MI 48109, USA. [5] Division of Biological Sciences, University of California, San Diego, 9500 Gilman Drive MC 0380, La Jolla, CA 92093, USA. [6] Department of Mechanical Engineering, University of Michigan, 2350 Hayward Street, Ann Arbor, MI 48109, USA. [7] Robotics Program, University of Michigan, Ann Arbor, MI 48109, USA. Correspondence and requests for materials should be addressed to T.Y.M. (email: taliaym@gmail.com)

Locomotion is an essential tool in the evolutionary "arms race" between predator and prey. Convergent locomotor behavior in prey species can indicate a broadly successful evasion strategy, and the success of a particular evasive maneuver can be measured by directly observing predator–prey interactions[1]. However, understanding the aspects of locomotion that enhance predator evasion ability makes it possible to predict prey success in broader contexts, especially where direct observation of predator-prey interaction may not be feasible.

The evasive success of prey locomotion can be defined in the context of the predation strategy encountered[2]. This mechanistic understanding of predator–prey interactions enables the inference of predator evasion ability, even without direct observation of predation events. For example, maintaining high uniform velocities can be useful for prey that are hunted by predators using a simple pursuit strategy[3]. Often, the successful animal in a pursuit is the one with greater speed or endurance. On the other hand, predation based on ballistic interception requires the prediction of prey movement to plan a predator strike. Therefore, increasing the unpredictability of prey trajectories likely increases the chance of evading a predator's ballistic interception[2, 4, 5]. Although the notion of unpredictability has been formalized in the field of Information Theory, this concept has not yet been applied to quantitatively characterize animal locomotion. Rather, previous studies of prey escape trajectories are limited to qualitative descriptions of "evasive maneuvers"[6–8] or measurements of variance in speed or direction[9–11], neither of which is a comprehensive measure of unpredictability. We present the first quantitative method to measure the unpredictability of motion in three-dimensional space by calculating the differential entropy of animal trajectories.

We demonstrate the utility of this method by examining whether bipedal locomotion increases predator evasion ability in desert rodents via increasing trajectory unpredictability. Transient bipedal locomotion is often associated with prey escape trajectories, and is performed by a variety of terrestrial animals, including lizards and cockroaches when chased, to achieve high running speeds[12]. In contrast, obligate bipedal locomotion has convergently evolved in desert rodents that are hunted via ballistic interception by owls and snakes[5, 6, 13]. Previous research has suggested that bipedal locomotion increases predator evasion ability with respect to sympatric quadrupedal rodents[14, 15]. However, the mechanism by which bipedalism increases predator evasion ability has not been identified.

Here we evaluate the kinematic, dynamic, and behavioral changes associated with the evolution of bipedalism in rodents by comparing the locomotion of sympatric bipedal jerboas and quadrupedal jirds. Jerboas (family Dipodidae) are derived bipedal desert rodents[16], with elongate hindlimbs, three bipedal gaits, and erratic "ricochetal" locomotion that is often assumed to enhance evasion ability[17, 18]. Jirds (genus *Meriones*) are sympatric with jerboas and are quadrupedal, like the majority of rodents, including the ancestors of jerboas[16]. We find that jerboas significantly deviate from the gait usage patterns stereotypic of quadrupedal steady-state cursorial locomotion. Jerboas frequently transition between gaits with distinct dynamic functions, a behavior that likely contributes to increased maneuverability, which has the potential to enhance predator evasion ability. To test the hypothesis that bipedal jerboas have higher predator evasion ability than quadrupedal rodents, we measured the three-dimensional unpredictability of bipedal and quadrupedal rodent locomotion in response to simulated predation. Indeed, we found that bipedal jerboas increase their trajectory unpredictability with respect to sympatric quadrupedal jirds by increasing the likelihood of turning and leaping, likely increasing the jerboas' ability to evade ballistic interception predation.

Because exposed microhabitats are an important source of nutrient resources, there is a conflict in small foraging animals between exploration and risk of predation that determines how long an animal will stay in an open area[19]. Enhanced evasion ability decreases the risk of predation in exposed microhabitats, resulting in an inverse relationship between predator evasion ability and thigmotaxis—the behavioral affinity to shelter[19–21]. Thigmotaxis can therefore be used to indicate relative evasion ability between similar animals that encounter the same predators. We used standard assays of rodent behavior to test the prediction that bipedalism is associated with a decrease in thigmotaxis in jerboas, further supporting our hypothesis that bipedalism increases predator evasion ability. While this study is limited to one example of bipedalism in rodents, kangaroo rats and Australian hopping mice have similar biomechanical and ecological divergence from sympatric quadrupeds that may also be explained by the divergence in trajectory unpredictability we measured in jerboas[22–25].

Our study, which integrates laboratory- and field-based analyses of biomechanics and behavior, presents a new metric for quantitatively characterizing the entropy of non-steady-state locomotion that substantially broadens and contributes meaningfully to our understanding of interspecies interactions in a natural setting.

## Results

**Gait dynamics**. Although bipedalism has evolved multiple times in rodents, jerboas are the only group observed to employ three different footfall patterns, or gaits[18, 26, 27]. The jerboa's hopping gait is similar to kangaroo hopping — both hind limbs contact the substrate simultaneously and then have extended aerial phases, in which the animal is not in contact with the ground. Skipping involves staggered but overlapping hind limb contact with the ground interposed between aerial phases. Finally, jerboa running is similar to human running, with an aerial phase following the contact of each hind limb with the ground.

Animals with multiple terrestrial gaits are often specialized for sustained locomotion at high speeds, frequently described as cursorial. As the speed of locomotion increases, cursorial animals transition between gaits to minimize the cost of transport and reduce the loading impact on the musculoskeletal system[28, 29]. A low cost of transport increases the endurance of cursorial animals, enhancing their performance while migrating, outrunning predators, or chasing down prey[30]. Thus, patterns of energy consumption can potentially indicate the selective pressures shaping the evolution of locomotion. Similarly, previous observations suggest that jerboas hop at the lowest speeds, skip at intermediate speeds, and run at the highest speeds[17, 18].

We first examined whether each jerboa gait is used exclusively at the speed range expected for cursorial locomotion. The speeds at which cursorial gaits occur can be predicted by the ratio of centripetal force to gravitational force (as an animal moves over its supporting limb), or the Froude number[31]. Given a leg length of 0.061 m (mean hip height at mid stance), dynamic similarity based on equivalent Froude numbers predicts that hopping should occur predominantly below $0.54\,\mathrm{ms^{-1}}$, skipping should occur predominantly from 0.54 to $1.21\,\mathrm{ms^{-1}}$, and running should occur predominantly at speeds above $1.21\,\mathrm{ms^{-1}}$. We found no locomotion under $0.5\,\mathrm{ms^{-1}}$, and surprisingly found no significant difference between the mean speed of each gait ($F_{2,77} = 2.82$, $P = 0.07$, one-way ANOVA, Fig. 1a), though the highest speeds (up to $3.02\,\mathrm{ms^{-1}}$) were observed during hopping and the lowest speeds during running. Skipping exhibited the greatest range of speeds, and was used most often (64 of 80 single-gait trials). Although the maximum speed exhibited by jerboas in the

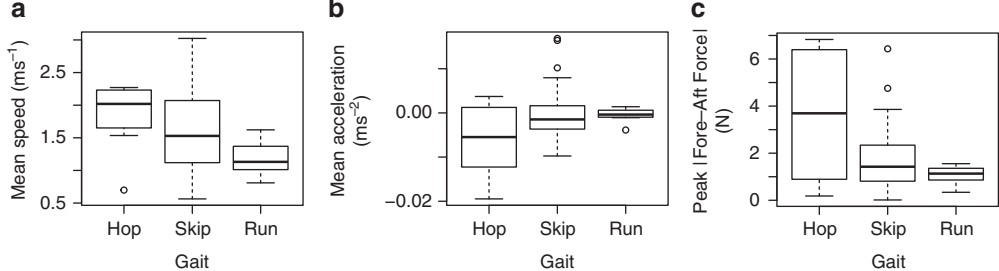

**Fig. 1** Jerboa gaits are not associated with the distinct speed ranges expected for cursorial locomotion. Boxplots showing **a** mean speed (magnitude of total x- and z-direction velocity), **b** mean acceleration (derivative of speed in **a**), and **c** the peak absolute value of fore-aft force recorded by the force platform over n = 80 trials. Mean speed is not as strong a predictor of gait ($F_{2,77}$ = 2.82, P = 0.07, one-way ANOVA) in *J. jaculus* as mean acceleration ($F_{2,77}$ = 3.99, P = 0.02, one-way ANOVA). The *boxes* span the interquartile range, the *bold line* represents the median, the whiskers extend to 1.5 times the interquartile range, and the *open circles* show outlier values outside of the whiskers

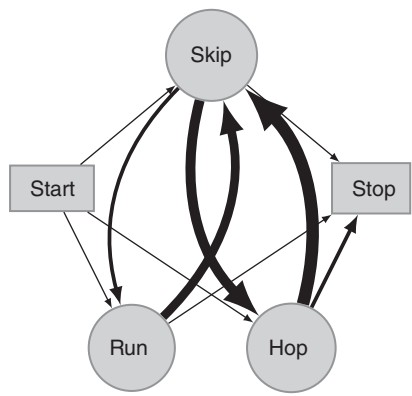

**Fig. 2** Path diagram of observed transitions between gaits, starting, and stopping for 36 trials. The path thickness represents the frequency of transitioning from one gait to another. The camera field of view included the center of the track, not the ends, resulting in a bias towards capturing more stopping, rather than starting, events

**Table 1 ANOVA table showing *F*-statistic and *p*-value for gait as a predictor of locomotor forces**

| Vertical Force | Peak F, p | \|Peak\| F, p | Mean F, p | Minimum F, p |
|---|---|---|---|---|
| Gait | 2.513, 0.088 | 2.513, 0.088 | 0.003, 0.997 | 1.426, 0.246 |

| Fore-Aft Force | Peak F, p | \|Peak\| F, p | Mean F, p | Minimum F, p |
|---|---|---|---|---|
| Gait | 0.222, 0.801 | 9.32, 2.37e − 4* | 2.055, 0.135 | 7.097, 0.015* |

All tests had 2 degrees of freedom. Out of 80 trials, 64 were "Skip," 8 were "Hop," and 8 were "Run." Asterisks indicate statistical significance ($\alpha$ = 0.05). \|Peak\| is the maximum of the absolute value of force values throughout a trial

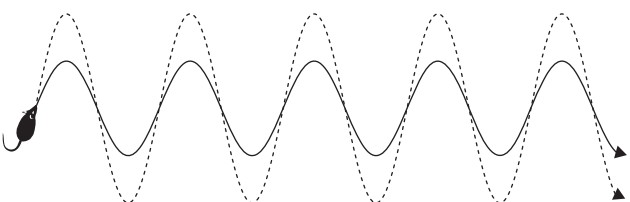

**Fig. 3** Diagram showing two sinusoidal trajectories from a bird's eye view, traveling from *left* to *right*. The *dashed* path has twice the amplitude of the *solid* path. Consequently, the *solid* path has twice the variance of the *dashed* path, though both paths have the same unpredictability (H)

laboratory is lower than the maximum speeds jerboas may exhibit in the field[17], the maximum speed exhibited in the laboratory exceeds the speed at which the Froude equation predicts a transition to a running gait. Furthermore, the absence of locomotion at the lowest speed range predicted by the Froude equation, the substantial overlap between the speed ranges of each gait, and the fact that all three gaits were observed at submaximal speeds, contradict the expectation based on cursorial locomotion that speed regulates gait usage in jerboas.

We next quantified acceleration and ground-reaction forces to examine the dynamics underlying each gait. We found significant differences in mean acceleration among the gaits ($F_{2,77}$ = 4.00, P = 0.02, one-way ANOVA, Fig. 1b). Hopping was associated with the greatest values of acceleration and deceleration (Supplementary Movie 1). Indeed, a path analysis of transitions between gaits showed that hopping was preferentially used to decelerate, or stop locomotion (Fig. 2). Skipping exhibited the most symmetrical variation in mean acceleration across the broadest range of speeds (Supplementary Movie 2), and was the gait to which animals transitioned most frequently (Fig. 2). Running showed the least variation in mean acceleration, and was used primarily at lower speeds (Supplementary Movie 3). The ground-reaction force data confirmed and explained these trends by revealing that gait is a strong predictor of both minimum fore-aft (or decelerative) and the absolute value (or magnitude) of fore-aft force (Table 1, Fig. 1c, Supplementary Fig. 1). Furthermore, ground-reaction forces did not show a

characteristic pattern through time for each gait, as would be expected for a cursorial animal using steady-state locomotion, and the vertical forces were not significantly different between gaits (Supplementary Fig. 2). These results suggest that jerboas transition frequently between gaits that exhibit distinct dynamic functions, thus increasing their capacity for maneuverability (Supplementary Movie 4).

**Trajectory unpredictability**. While morphology determines the "capacity" to generate complex behaviors (i.e., maneuverability), this only defines the limits of theoretical performance[32, 33]. Predator–prey interactions are determined by the "observed" performance of locomotor trajectories that result from path planning behavior, regardless of the animal's theoretical maneuverability[2]. This distinction is explicitly measured by computing the unpredictability, or entropy, of the realized

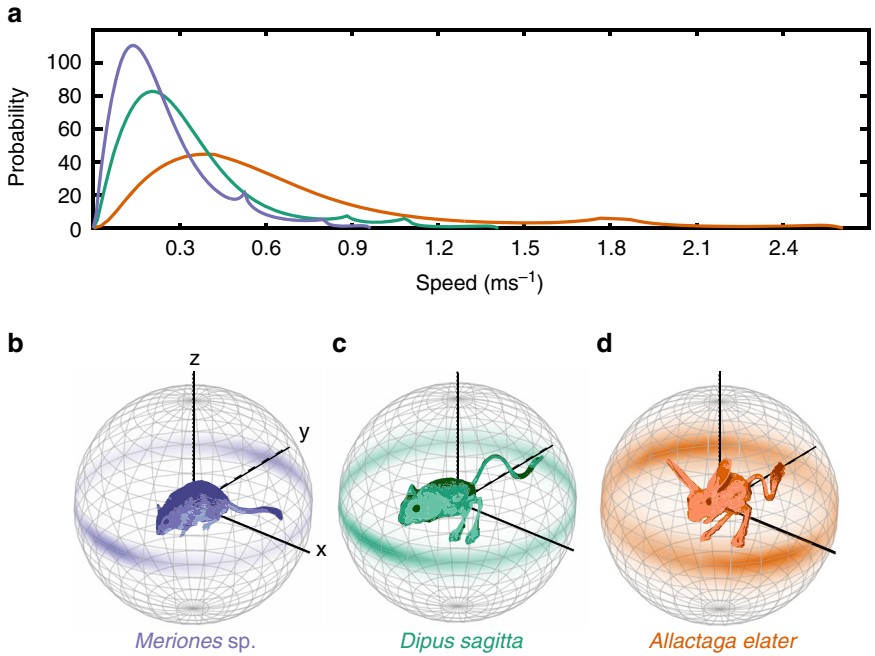

**Fig. 4** Differences in unpredictability between species result from species-specific patterns of speed and direction of motion. Continuous probability distributions of **a** speed and **b–d** angle of locomotion for field animals. In all subplots, the quadrupedal jird, *Meriones sp.*, is shown in *blue*, the bipedal jerboa, *Dipus sagitta*, is shown in *green*, and the bipedal jerboa, *Allactaga elater*, is shown in *orange*. In **a**, the integral of each probability distribution is equal to one. In **b–d**, the color opacity at a given angle on the sphere corresponds to the probability of moving at that angle, with respect to the animal location in the previous frame (the center of the sphere). **b** Shows the probability distribution of the quadrupedal jird, *Meriones sp.*, movement angle in *blue*, **c** shows the probability distribution of the bipedal jerboa, *Dipus sagitta*, movement angle in *green*, **d** shows the probability distribution of the bipedal jerboa, *Allactaga elater*, movement angle in *orange*. Spheres are symmetric about the equator due to the symmetry of the ascent and descent about the apex of a leap. Asymmetry about the y axis in *Allactga elater* is likely the result of asymmetry in the predation stimulus

performance of locomotor trajectories:

$$H(p) = -\sum_{i \in X} p(i) \log p(i) \qquad (1)$$

where $H$ represents entropy, $X$ is the set of possible outcomes, or the set of three-dimensional positions relative to an animal's current position, and $p(i)$ is the probability of performing action $i$ in $X$, or the probability of the animal occupying a position $i$ in three-dimensional space relative to its current position[34]. Morphological traits that enable bipedal rodents to move extensively in the vertical direction increase the number of locomotor possibilities (increasing $X$) available to the animal, but the animal must "decide" to use these specialized morphological traits to generate more unpredictable locomotor trajectories (higher $H$).

The variance of observed escape trajectories has been used in previous studies to estimate predator evasion ability (reviewed in ref. [9]), but unpredictability more accurately estimates a predator's inability to compute a successful intercept course[35]. To illustrate, consider a prey animal with a sinusoidal trajectory (Fig. 3). The animal can increase its variability by increasing the amplitude of the lateral displacements, yet the overall pattern, and thus the unpredictability, of the locomotion remains the same. To quantify the effect of bipedal locomotion on trajectory unpredictability, we therefore developed methods to directly measure the unpredictability of rodent locomotion by using the notion of differential entropy that generalizes the entropy Eq. (1), below, from discrete to continuous probability distributions.

In an effort to quantify the effect of bipedalism on trajectory unpredictability, we observed the locomotion of freely moving wild-caught sympatric bipedal and quadrupedal rodents in response to simulated predation in their natural habitat

(Supplementary Movies 5 and 6). We captured and observed two species of bipedal jerboa (*Allactaga elater* and *Dipus sagitta*) and one species of quadrupedal jird (genus *Meriones*). *D. sagitta* is a three-toed jerboa closely related, and of the same morphotype, as the *J. jaculus* jerboas in our laboratory experiments, and *A. elater* is a bipedal five-toed species that only makes contact with the ground via their three central hindlimb digits[16]. As expected, bipedal jerboa trajectories were more unpredictable (higher entropy) than quadrupedal jird trajectories ($H(A.\ elater) = -9.936 > H(D.\ sagitta) = -12.60 > H(Meriones\ sp.) = -13.91$). When each is compared to a null-hypothesis generated from a mixed data set selected from each pair of species (See Supplementary Methods for a description of this modified $t$-test), significant differences were found between the entropy (and therefore unpredictability) of bipedal *A. elater* and quadrupedal *Meriones sp.* (Supplementary Fig. 3A, *A. elater* $P = 0$, *Meriones* sp. $P = 0.0002$, modified t-test). However, we were surprised to find that the difference in trajectory entropy between the bipedal jerboa *D. sagitta* and quadrupedal jird *Meriones* sp. was only trending towards significant (Supplementary Fig. 3C, *D. sagitta* $P = 0.0561$, *Meriones* sp. $P = 0.0605$, modified t-test) at the $\alpha = 0.05$ level. We also found that the two bipedal jerboa distributions differed significantly (Supplementary Fig. 3B, *A. elater* $P = 0$, *D. sagitta* $P = 0.0002$, modified $t$-test). Thus, each sympatric species moved with a distinct pattern of trajectory unpredictability.

To identify the locomotor behaviors that contribute to unpredictability in each species, we projected the multidimensional probability distribution into two marginal distributions: one illustrating the distribution of speeds (Fig. 4a) and another illustrating the distribution of angles (Fig. 4b–d). Quadrupedal jirds (*Meriones* sp.) used lower speeds most frequently, and the

bipedal jerboa *A. elater* exhibited the most even distribution of speeds, reflecting a greater probability of acceleration and deceleration from step to step (Fig. 4a). The distributions of angles of motion reveal spatial preferences for each species. Despite more frequent contact with the ground, and therefore more opportunities to change direction in the horizontal plane, quadrupedal jirds showed the greatest preference for forward locomotion with few turns (Fig. 4b). In contrast, both jerboa species showed an increased preference for vertical motion (indicated by the longitudinal extent of the colored clouds in Fig. 4c, d), suggesting that the evolution of bipedalism likely facilitated the divergence in trajectory unpredictability between jerboas and jirds by allowing jerboas to utilize three-dimensional space and increasing their capacity for acceleration. Differences in entropy between the two jerboa species resulted primarily from the stronger preference of *A. elater* for turning (indicated by the extent of the colored clouds away from the *y* axis in Fig. 4c, d), and to a lesser extent a difference in the propensity for vertical motion. The difference in preference of horizontal movement direction explains how two species of bipedal jerboa can exhibit such significant differences in locomotor unpredictability.

**Risk aversive behavior.** Small, foraging animals that are susceptible to predation are subject to a conflict between the desire to explore new areas to discover food and the desire to remain in covered areas safe from predators. Due to this conflict, the amount of time an animal spends in an open area varies with their predator evasion ability (see Eq. (1)[19]), assuming that there is a minimum acceptable probability of death. Therefore, to determine whether prey with unpredictable trajectories have greater predator evasion ability, we measured rodent thigmotaxis.

During the simulated-predation trials we observed that the bipedal species explored the entire arena, while quadrupedal jirds had an apparent affinity for the walled periphery (Supplementary Movies 5 and 6). We therefore retrospectively analyzed the same locomotor trajectories according to the Open-Field Exploration Test, frequently used to assess thigmotaxis in rodents[36]. This test interprets a decrease in exploration and general locomotor

activity in a brightly lit and exposed environment as a stress response to being without refuge from predators. Despite the stress being provided by simulated-predation rather than the more conventional bright lights, we found a significant increase in the percentage of time bipedal species spent in exposed areas (Fig. 5a).

To determine whether the divergence in microhabitat preference could be replicated under more controlled conditions, we performed the Light-Dark Box Exploration test, a standard assay of thigmotaxis with unambiguous categories of behavior and minimal restriction on locomotion[36]. We recorded the amount of time each animal spent in either the protection of a dark box or in exposed and brightly-lit open areas (Supplementary Movies 7 and 8). Lab-reared bipedal jerboas (*J. jaculus*) spent significantly more time exploring the open area than the lab-reared quadrupedal jirds (*Meriones unguiculatus*) (Fig. 5b, Welch Two Sample *t*-test: $P = 0.005$, $t = -3.4113$, $df = 11.756$, Supplementary Table 1). These results corroborate the pattern we found in the natural habitat and suggest bipedalism is associated with less thigmotaxis and greater predator evasion ability in open microhabitats.

## Discussion

While laboratory-based studies of steady-state locomotion establish functional relationships between morphology and motion, these studies are highly abstracted from the way animals behave in natural circumstances[11, 37]. In predator–prey interactions, prey survival is often determined by momentary bursts of acceleration and rapid evasive maneuvers[1, 19]. Creating a tool that specifically characterizes these types of transient, non-steady-state locomotion makes it possible to more closely relate biomechanical performance to evolutionary fitness.

In general, non-steady-state locomotion enables prey to evade predators that hunt using a pre-calculated intercept course[2, 5, 8]. Ricochetal bipedal locomotion in rodents may be an adaptation to enhance predator evasion[18, 38]. For example, Australian hopping mice transition from quadrupedal to bipedal locomotion during evasive maneuvers[27]. Kangaroo rats are obligately bipedal, produce evasive maneuvers in response to predator cues, and have greater evasion success than sympatric quadrupedal rodents[6, 7, 39]. To determine whether the shift to bipedal locomotion enhances predator evasion, we developed metrics for quantifying the evasiveness of a maneuver. Since bipedal locomotion with large aerial phases increases the dimensionality of rodent locomotion, we sought out analytical methods capable of quantifying the contributions of both velocity and dimensionality to identify differences observed between bipedal and quadrupedal rodent locomotion.

Highly variable animal behavior is often characterized with ethograms to record the frequency and variety of behavioral components[40, 41]. Unfortunately, these methods require discretization of continuously varying behavior, which introduces bias into the identification of behavioral components[42]. As we have shown, jerboas move with step-to-step changes in stride length, direction, gait, and speed, making it difficult to determine an appropriate discretization method. Sorting jerboa locomotion by each of these unique factors would require an ethogram with an infinite number of states or a binning that is so coarse that it obscures underlying patterns of behavior.

Instead, to quantify the unpredictability of locomotion that is continuously varying in both space and time, we developed a method to measure the differential entropy of locomotor trajectories. In addition to a quantitative metric of unpredictability that corresponds to predator evasion ability, an initial step in calculating entropy is constructing a continuous model of the animal's locomotion. When interpreted together, the model

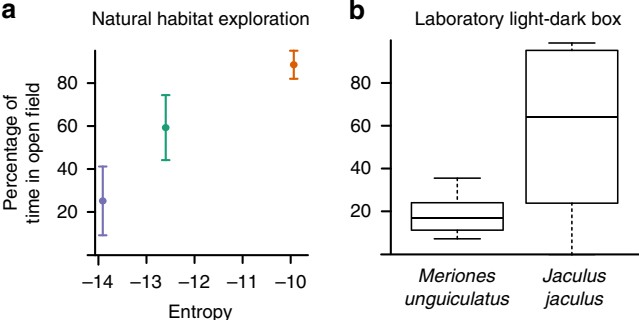

**Fig. 5** Bipedal, and more unpredictable, locomotion is associated with lower open-field anxiety in sympatric desert rodents. **a** Percentage of observed time spent in open field by species, as a function of the entropy of each species' trajectories during simulated predation trials in the natural habitat enclosures. The quadrupedal jird, *Meriones* sp., is shown in *blue*, the bipedal jerboa, *Dipus sagitta*, is shown in *green*, and the bipedal jerboa, *Allactaga elater*, is shown in *orange*. Error bars indicate 95% confidence intervals for each species (*n* (*Allactaga elater*) = 14 videos, *n* (*Dipus sagitta*) = 6 videos, *n* (*Meriones* sp.) = 5 videos). **b** Percentage of time spent in open field is significantly different between species in laboratory Light-Dark Box trials (Welch Two Sample *t*-test: P = 0.005, t = -3.4113, df = 11.756, 12 *J. jaculus* trials, 18 *M. unguiculatus* trials). The boxes span the interquartile range, the *bold line* represents the median, the whiskers extend to 1.5 times the interquartile range

explains the components of movement that contribute to the animal's overall trajectory unpredictability.

Many existing metrics of characterizing terrestrial motion use animal morphology as the frame of reference, often standardizing by body size to compare the theoretical relationship between morphology and performance between multiple individuals of the same species, or multiple dynamically similar species. In the context of predator–prey interactions, defining motion with respect to prey morphology is less relevant to prey fitness than defining motion from the perspective of the predator. Because our entropy calculation is agnostic to the theoretical capabilities of an animal's morphology, it can quantitatively compare motion between animals that move with different mean stride lengths and in different planes of locomotion. Just as many predators are capable of hunting multiple species of prey, our entropy method enables relevant comparison between sympatric species that encounter similar selective pressures.

By revealing the biomechanical principles underlying inter-species interactions, the entropy method has the potential to explain the ecological structure of a community. For example, in desert ecosystems, small quadrupedal rodents tend to forage near bushes (high thigmotaxis), where the risk of predation is low and the likelihood of finding seeds is high[43–46]. This behavioral aversion to exposure is so ingrained in mice that it is a standard experimental system for understanding the genetics of anxiety and for measuring the effectiveness of therapeutic pharmaceuticals[36]. The increased trajectory unpredictability in bipedal jerboas enhances predator evasion and therefore decreases the risk associated with foraging in the exposed microhabitats common to sparsely vegetated deserts. Furthermore, competition with quadrupedal rodents would decrease if jerboas evolved a behavioral propensity to forage in exposed areas, where quadrupedal rodents are at higher risk of predation.

On the basis of the relationship we found between trajectory unpredictability and microhabitat preference, the evolution of bipedalism has the potential to influence desert community ecology. It is clear that divergence in microhabitat preference can enable resource partitioning to decrease competition in resource-poor ecosystems[14, 47]. The substantial differences we have recorded in trajectory unpredictability and open-field anxiety between sympatric jerboas and jirds (Fig. 5), along with several studies noting similar biomechanical and ecological divergence between bipedal and quadrupedal rodents on other continents[22–25], suggest that bipedalism may have evolved to limit interspecific competition, thereby maintaining diversity among desert rodents[14, 15]. Indeed, many communities of desert rodents in Africa[18, 48] and North America[20, 49, 50], exhibit similar patterns of locomotor, behavioral, and ecological stratification that include the evolution of obligate bipedalism, whereas South American desert communities without bipedal rodents are less taxonomically diverse[51]. Thus, since diversity in predator evasion ability and foraging strategy can enable resource partitioning and species coexistence[52, 53], we propose that locomotor innovation is an important factor that should be examined for its influence on the taxonomic diversity of animal communities.

This analysis of predator evasion ability in desert rodents demonstrates how our computational framework can be used to integrate biomechanical studies with other fields of biology. In addition to predicting prey fitness, this metric can be applied to quantitatively assess a variety of important motion behaviors. Measuring the entropy of movements through time can reveal how motor control is learned ontogenetically or evolutionarily[54, 55]. Just as the complexity of a bird song is an important signal to female songbirds, the complexity of behavioral mating displays can now be tested for its effect on fitness[56, 57]. Furthermore, combining entropy-based measurements of behavioral models

and mimics with phylogenetic comparative methods can reveal the pattern and process of evolving behavioral mimicry[58]. Thus, the framework we present for characterizing non-steady-state locomotion encourages an integrative biomechanical approach that complements and mechanistically informs behavior, ecology, and evolution.

## Methods

**Animal care and use**. All animal care and use protocols were approved by the Harvard Faculty of Arts and Sciences Institutional Animal Care and Use Committee (IACUC protocols 28–23, 20–09) and the United States Department of Agriculture.

**Laboratory gait analysis**. We collected 139 trials from 5 male jerboas (*J. jaculus*) from a breeding colony at the Harvard Concord Field Station. Each animal was encouraged to travel along a narrow trackway ($2 \times 0.15 \times 0.4$ m$^3$) over a two-axis force platform ($0.06 \times 0.12$ m$^2$) and past a high-speed video camera recording at 500 fps. The camera field of view captured one to three strides, depending on speed. The size of the trackway had no detectable effect on the range of speeds used by jerboas when compared to a previous study of unrestricted jerboa locomotion indoors (0.5 to 3.21 ms$^{-1}$)[18].

We visually categorized the gait of each stride by footfall pattern following terminology specific to bipedal animals[12], rather than terminology traditionally used to describe quadrupedal gaits[18]. A total of 59 trials were excluded from speed and force analysis: 28 involved transitions between gaits, 8 included a stop or start from standstill (these 36 trials were used to generate Fig. 2), 2 involved changes between the leading foot between strides, and the gait in 20 trials was ambiguous. The remaining data set included 80 trials, with 7–29 from a single individual.

We calculated the speed of each jerboa by tracking its eye using DLTdv5 tracking software for Matlab[59]. The eye was used to estimate the center of mass, as the *J. jaculus* have fused cervical vertebrae[16, 60], which limit head motion with respect to the body.

Gait transition predictions were made assuming that the transition from slow-to-medium gaits occurs at $Fr = 0.5$, and the transition from medium to fast gaits occurs at $Fr = 2.5$, and using the equation

$$Fr = \frac{u^2}{gl} \qquad (2)$$

where $l$ is the leg length, $g$ is gravity, and speed is $u$[31]. Leg length was estimated as the height of the hip at mid-stance for each trial. For $n = 80$ trials, the mean leg length was 0.0605 m, 0.0066 m standard deviation.

**Field trajectory unpredictability**. We captured rodents from their natural habitat in the desert north of Fukang, in Xinjiang Province, People's Republic of China. Bipedal jerboas (*A. elater* and *D. sagitta*) and quadrupedal jirds (*Meriones* sp.) were captured at night, then observed and released the following night. Animals were recorded moving within a $5 \times 5$ m$^2$ fenced area in the natural habitat during their naturally active hours (21:00–1:00 CST, 19:00–23:00 Xinjiang Local Time). We illuminated the enclosure using infrared floodlights and recorded animal movements using cameras from which the infrared filter was removed (Casio ZR100). Each animal was filmed in HD ($1920 \times 1080$ pixels) at 30 fps by two cameras as it responded to stimuli (e.g., looming, loud noises) for 15 min or until it presented signs of fatigue. In light of humans being slower and less maneuverable than the rodents' natural predators, each trial consisted of one human stationed at each end of the enclosure. This had the effect of scaring the rodent towards the other human, increasing the rapidity of each encounter. The asymmetry in movement preference about the forward direction in *A. elater* (Fig. 4b) was likely in response to asymmetry in the human-generated stimulus, as random pairs of four individuals produced the stimulus from opposite ends of the enclosure, and one individual often provided a more effective stimulus than the other.

The center of the rodent body was tracked in three-dimensional space using DLTdv5 tracking software[59]. The calibrated volume was $4.3 \times 2.5 \times 0.4$ m$^3$. In each video, continuous segments of the tracked data < 20 frames were excluded, and each remaining segment of the continuously tracked data is hereafter referred to as a "trial." To reduce the effect of outliers, we retained only the most similar 97% of the data for each species. This data set included 25,397 frames for 5 *A. elater*, 11,185 frames for 3 individuals of *D. sagitta*, and 20,609 frames for 3 individuals of *Meriones* sp. We chose to use kinematics tracking rather than other tracking devices (e.g., radio-frequency ID or PIT tags), which do not operate well over longer ranges and do not provide information regarding three-dimensional movement patterns.

**Entropy calculation**. To compare the locomotor unpredictability of each species, we developed methods to measure the entropy of trajectories in continuous three-dimensional space. Previous methods of entropy calculation require that data be either discretized and organized into a histogram or that the data be in the form of a continuous distribution. While our data are finite, we sought to avoid arbitrary

discretization of three-dimensional space. Therefore, we used optimization to find the continuous probability distribution that best fit our finite data while making the fewest assumptions about the shape of the data. Measuring the entropy of the continuous probability distribution that best fits the observed data provides the unpredictability of the trajectories performed by each species. The mathematical steps leading to the optimization problem are presented below. The algorithmic solution to the optimization problem and the necessary mathematical proofs are presented in the Supplementary Methods.

To measure the entropy of a species, we first determined the probability with which each species travels to any given location in a given interval of time. For example, if the species prefers to move forward on the horizontal plane at high speeds, the animal would have a high probability of occupying locations directly in front and far from the present location; all other locations would be associated with a low probability. Since three-dimensional space and speed are continuous variables, the probability of occupying any potential location, given the current location, is described by a continuous probability distribution. Because our data are finite, we assume that they are sampled from a continuous probability distribution that represents the true movement preferences of each species. We seek to measure the unpredictability of each species' true movement preferences, therefore we measure the entropy, $H[f]$, of the continuous probability distribution, $f$, that best fits the finite data collected for each species, $d$.

The entropy of continuous probability distributions is called "differential entropy." Given a continuous probability density function, with distribution $f$, the differential entropy is defined as:

$$H[f] = - \int_{[-1,1]^3} f(\mathbf{x}) \ln f(\mathbf{x}) d\mathbf{x}. \tag{3}$$

where the limits of the integral are defined by the maximum and minimum distance that the animal can move in any direction in an instance of time, and $\mathbf{x}$ is a point in three-dimensional space. Note, in this case we have chosen −1 and 1 for convenience in the presented equations, but these values represent the minimum and maximum distance an animal can travel in three-dimensional space within one frame. Negative distances in this case correspond to backwards movement with respect to the previous frame.

However, constructing a continuous probability distribution that best explains the finite observations is non-trivial because a finite amount of data can be explained by an infinite number of continuous probability distributions. A continuous probability distribution is said to "fit" a finite set of observed data if the moments of the distribution match the moments of the data[61]. Statistical moments are quantitative measures that describe the shape of a data set. For example, the first moment, $m_1$, of a scalar dataset is the mean, the second moment, $m_2$, of a scalar dataset corresponds to variance, etc.

The moments, $\mathbf{m_\alpha}$, for a continuous probability density function, $f$, are defined as:

$$\mathbf{m_\alpha} = \int_{[-1,1]^3} \mathbf{x^\alpha} f(\mathbf{x}) d\mathbf{x}, \tag{4}$$

where $\mathbf{x} = (\mathbf{x}_1, \mathbf{x}_2, \mathbf{x}_3) \in \mathbb{R}^3$, $\mathbb{R}^3$ denotes the set of three dimensional real numbers, $\mathbf{x^\alpha}$ denotes $\mathbf{x}_1^{\alpha_1} \mathbf{x}_2^{\alpha_2} \mathbf{x}_3^{\alpha_3}$ with $\alpha_1, \alpha_2, \alpha_3 \in \mathbb{N}$, and $\mathbb{N}$ denotes the set of nonnegative integers. Moving forward, we write $(\alpha_1, \alpha_2, \alpha_3) \in \mathbb{N}^3$ where $\mathbb{N}^3$ denotes the three-dimensional set of nonnegative integers.

On the other hand, empirical moments, $\widehat{\mathbf{m}}_\alpha$, describe the moments of finite observations. Suppose we are given individual data points $\mathbf{d}_1, ..., \mathbf{d}_N$ where $\mathbf{d}_i \in \mathbb{R}^3$ for all $i \in \{1, ..., N\}$. Then the empirical moments, $\hat{\mathbf{m}}_\alpha$, of the data are defined as:

$$\widehat{\mathbf{m}}_\alpha = \frac{1}{N} \sum_{i=1}^{N} (\mathbf{d}_i)^\alpha. \tag{5}$$

Under mild assumptions, with an infinite set of empirical moments, a continuous probability distribution could be selected that perfectly matches the observed data[61]. However, computing an infinite number of moments is practically infeasible. A finite number of moments results in many possible continuous probability distributions that fit the empirical data. While making assumptions regarding the type of distribution producing the observed data (e.g., Gaussian) would reduce the number of possible continuous probability distributions, we have no basis for making such assumptions. To make the fewest assumptions about the type of distribution producing the observed data ("epistemic modesty"), we follow the Principle of Maximum Entropy[62], which states that, amongst all the continuous probability distributions that fit the data, the best distribution is the one that maximizes the entropy. Therefore, we seek the continuous probability distribution that maximizes the entropy while matching the empirical moments of the observed data.

We can therefore restate this as an optimization problem:

$$\max_{f} \left\{ H[f] \,\middle|\, \int_{[-1,1]^3} \mathbf{x^\alpha} f(\mathbf{x}) d\mathbf{x} = \widehat{\mathbf{m}}_\alpha, \, \forall \alpha \in \mathbb{N}_{2k}^3 \right\}, \tag{6}$$

where $\mathbb{N}_{2k}^3$ refer to those $\alpha \in \mathbb{N}^3$ with $|\alpha| = \sum_{i=1}^3 \alpha_i \leq 2k$ (see Supplementary Methods). That is, we seek to find a continuous probability density function, $f$, that simultaneously maximizes the differential entropy while having its first $2k$ moments equal to the given empirical moments (see Algorithm 1 in Supplementary Methods to solve this optimization problem). Once the continuous probability function that solves this optimization problem is found, the differential entropy of the distribution can be measured. Furthermore, the continuous probability function can be visualized by examining projections of the distribution with respect to different variables (e.g., speed or angle of motion in three-dimensional space).

**Field-based open-field exploration test.** The same videos used to capture field trajectory unpredictability were visually analyzed for field usage following the "Open Field Exploration Test" as described in ref. [36]. Animals within two body-lengths of the enclosure walls were categorized as "edge," including climbing on the walls of the enclosure. All other times the animals were visible were categorized as "open."

**Laboratory-based light-dark box exploration test.** We used four adult male *J. jaculus* jerboas born (in March 2013) and raised in the laboratory colony at the Concord Field Station mentioned above. 6 outbred adult gerbils (*M. unguiculatus*) were obtained from Charles River, had 3 weeks to acclimate, and were 70–72 days old at the time of experimentation. All rodents were housed individually, and animal housing, care, and experimental protocols were approved by Harvard IACUC and USDA. Each animal performed three trials. The location of the dark box and the animal trial order were randomized prior to experimentation. The trials were performed throughout the periods in which their housing rooms were dark and illuminated (light cycle).

We performed this anxiety test following the procedures in ref. [36]. Animals were transported from their home cages in an opaque plastic transportation case that was sanitized between trials. We recorded the amount of time it took to capture each animal prior to experimentation to account for the stress of capture. Video was recorded from above with the same camera as in the simulated predation (Casio ZR100) in HD (1920 × 1080 pixels) at 30 fps. Recording began before the animal was brought into the experimental area in a sterilized plastic box, and ended after the animal was removed from the experimental arena. The experimental arena (1 × 1 × 0.35 m³) was made from clear plexiglas to aid in sterilization, with a yoga mat lining the bottom to provide grip. The shelter was an opaque plastic box (0.5 × 0.4 × 0.35 m³) with a 5 cm radius semicircular opening. The experimental arena was lit from above by a combination of fluorescent and incandescent lights, at 9000 lx across the exposed area. Animals were placed inside the experimental arena in the exposed area facing away from the shelter. The experimenter left the room for 5 min, retrieved the animal, and returned it to its cage. The experimental area, shelter, and transport box were sterilized with a disinfectant (Quatricide) between trials to limit the possibility of scent signals from a previous animal affecting the behavior of other animals. We visually recorded the amount of time each animal spent within the shelter vs. outside of the shelter, time spent in risk-assessment (face and forelegs outside of the shelter, with rest of the body within the shelter), and number of transitions between light, dark, and risk-assessment.

**Code availability.** Matlab code is available from the authors upon request.

**Data availability.** The data are available from the authors upon request.

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

## Acknowledgements

This study was funded in part by the Chapman Memorial Scholarship to T.Y.M. The authors thank S (Paine) Valdes, Y.C. Yin, J.F. Chu, K. Jayaram for experimental assistance, J.A. Miyamae for digitization, and P.A. Ramirez for animal care.

## Author contributions

All authors contributed to writing the text of the paper. T.Y.M.: Designed the study, performed experiments, and analyzed the data. K.L.C.: Designed the study and performed experiments. A.A.B.: Oversaw the experimental design. R.V.: Designed and performed the data analysis.

## Additional information

**Competing interests:** The authors declare no competing financial interests.

