## [Peer Review File · Nature Communications]

Reviewer #1 (Remarks to the Author):

This study characterizes “the unpredictability of non–steady–state locomotion, demonstrated by examining sympatric bipedal and quadrupedal rodent escape trajectories”. The method used to calculate the unpredictability of the subjects’ trajectories is novel and I agree with the Authors that unpredictability of motion is likely to be an important component of evading predators for many species. Unpredictability of escape trajectories may also be central to convergent evolution of bipedality in species of desert rodent species, as the Authors suggest. However, I have some concerns about some of the methods and I found the description of the calculation and analysis of locomotor entropy to be completely inadequate.

I like that the authors attempted to quantify differences in ground forces associated with the 3 gaits jerboas use to address the possibility that gait transitions are employed to increase unpredictability of trajectory. However, I am concerned that the way in which these data were collected renders the results largely useless. The 2 m long track way used to collect ground forces was too confining to allow a fair assessment of the locomotor behavior of these animals. Both the low speeds and accelerations (Fig. 1) suggest that the animals were seriously constrained. For this reason, I feel the collected data cannot provide fair evaluation of the degree to which the 3 gaits differ and may contribute to increased unpredictability of movement.

Also, in contrast to the assertion made in the abstract, it is not clear that this study demonstrates a link between gait transitions and unpredictability of locomotion.

I found the methods explaining the entropy calculation to be incomprehensible. The supplementary methods were equally obtuse. In the context of this study, what are “moments”. What empirical data was used and how was it used to calculate entropy. This section needs to be written in a way that even readers who may not be able to follow the math will be able to gain a sense of what was done.

Lines 101 – 114. The Authors state that they examined whether transitions between gaits are associated with speed ranges. But the range of speeds examined (Fig. 1) is less than half the maximum speed of jerboas. Without an analysis of close to the full range of speed, it is not possible to determine if particular gaits are associated with different ranges of speed.

Lines 162 – 164. The Authors state that they were surprised to find that *D. sagitta* was not significantly more erratic in its locomotion than the jird. The Authors should consider the extent to which the human “predator” actually stressed the jerboas. Compared to the predators actually face, humans are remarkably slow and lumbering. Consequently, the jerboa data may not represent maximum performance.

Lines 173 – 174. Please present the data discussed in this paragraph in tables or figures.

Assessment of field trajectory unpredictability, lines 299 – 314. How many cameras were used to document the 3D movements? At what framing speed were the videos recorded? How large was the calibrated volume in which analysis was done?

In Supplemental Figures 2 and 3, what are the units of force?

Supplemental Figure 4 should probably be included in the main body of the manuscript.

Reviewer #2 (Remarks to the Author):

This is an interesting study that explores the escape behaviour of bipedal and quadrupedal

mammals by developing a novel tool for quantifying the unpredictability of prey locomotor behaviour. The authors are correct in suggesting that this tool offers an opportunity for biomechanists to directly contribute to the field of ecology by linking movement behaviours to predator prey interactions and community dynamics.

The most important weakness of this manuscript is that this tool has not been tested in the field or the lab. The fundamental premise of the paper is that by quantifying the unpredictability of movement by animals then you will be able to provide an alternate metric of escape ability. This may be true but it does need to be tested – merely providing the logic for the metric and developing the math has only provided a first step. My feeling is that the authors should provide a rigorous test of the importance of locomotor unpredictability first before publishing, or the authors should publish this tool in a more specialized journal.

A potential shortcoming of the study is the limited number of species that have been observed. Comparing two bipedal species with just a single quadruped does not really provide us with the confidence that all bipedal species have more unpredictable locomotion than equivalent sized quadrupeds. This is especially important when one considers that there are such substantial differences between the two bipedal species that were studied.

One aspect of your manuscript that got me thinking is the relationship between the “increased” unpredictability of locomotion for the two bipeds and its possible link with their probability of being captured by predators. Your metric of entropy seems to be quantifying (I’m not absolutely sure of this, so this could be further explained) the probability that the animals will be in a certain location in the next time point given their current position. Clearly, the two bipeds can have greater changes in direction when in contact with the ground but they are also more likely to spend a greater amount of time in the air (flight time) when they absolutely cannot change directions. In contrast, the quadrupeds may be more predictable in their location at each time point, but they have more opportunities to change directions at any point in time. The quadrupeds may only be able to make small changes in direction when in contact with the ground, but they spend more time on the ground, thus allowing more opportunities for changes in direction. I’m then left wondering what these different locomotor attributes for the different species have for their probability of being captured by predators. This point reinforces to me the importance of testing the ‘benefits’ of your metric of locomotor unpredictability for escape success. I suspect this metric of entropy will be just one of many affecting escape success. Having said that, I don’t think this detracts from the novelty of your tool for studies of escape success but it does limit the novelty of this study when it is not tested empirically.

Reviewer #3 (Remarks to the Author):

The authors present a novel approach to assess behavior in the lab and in the field, and support conclusions regarding the evolution of bipedal hopping in desert rodents. In general, the paper is very well written and the rationale for the approach is clear. This work is both highly interesting and potentially very important. However, I do feel the authors overstate their conclusions somewhat in regards to the evolution of bipedal hopping. Specifically, as stated by the authors, jerboas are the most derived of the bipedal desert rodents. Therefore, it’s not clear how representative their behavior is of other species, or how well it explains the evolution of bipedality. While I do believe the approach taken by the authors provides interesting and novel insight into the behavior of jerboas, I would caution against overstating the role of unpredictability in the evolution of bipedal locomotion. For example, it is still likely that factors such as acceleration capacity also contributed to the evolution of bipedal hopping and may have in fact preceded the capacity for unpredictability.

I also have some minor points for clarification. Please see below.

In equation 2, how is leg length determined? This can vary greatly in species with a crouched posture.

You might consider using the square root of Fr (i.e., $u/(gl)^{0.5}$), as this eliminates the exponential. However, if the goal is to compare to older previously published data, this may not be necessary.

What feature on the animals was tracked in the field trajectory trials? If it was just the animal as a whole, how does position error effect the entropy calculation, i.e., how does digitizing a different position on the animal from frame to frame effect error?

Reviewer #1 (Remarks to the Author):

This study characterizes “the unpredictability of non–steady–state locomotion, demonstrated by examining sympatric bipedal and quadrupedal rodent escape trajectories”. The method used to calculate the unpredictability of the subjects’ trajectories is novel and I agree with the Authors that unpredictability of motion is likely to be an important component of evading predators for many species. Unpredictability of escape trajectories may also be central to convergent evolution of bipedality in species of desert rodent species, as the Authors suggest. However, I have some concerns about some of the methods and I found the description of the calculation and analysis of locomotor entropy to be completely inadequate.

I like that the authors attempted to quantify differences in ground forces associated with the 3 gaits jerboas use to address the possibility that gait transitions are employed to increase unpredictability of trajectory. However, I am concerned that the way in which these data were collected renders the results largely useless. The 2 m long track way used to collect ground forces was too confining to allow a fair assessment of the locomotor behavior of these animals. Both the low speeds and accelerations (Fig. 1) suggest that the animals were seriously constrained. For this reason, I feel the collected data cannot provide fair evaluation of the degree to which the 3 gaits differ and may contribute to increased unpredictability of movement.

We appreciate the reviewer’s concern that the trackway was too confining to elicit the full range of speeds. However, a previous study in an unrestricted laboratory area covered in sand and vegetation (Schroepfer et al. 1985) found speeds (0.5 to 3.21 m/s) that are comparable to the speeds we observed on our track (0.5 to 3.02 m/s) for the same species. Furthermore, Schroepfer et al. report that unconfined jerboas rarely perform more than a few jumps in a row, and jerboas in our trackway similarly stopped jumping after a few hops, often before they reached the force platform. Therefore, we would argue that a longer or wider track would likely have little to no effect on jerboa locomotion.

The confinement provided by the trackway was necessary to increase the probability that the jerboas would move over the force platform. This is a common constraint that most over-ground force platform studies require for wild animals. Although the trackway was confined mediolaterally, the vertical height of the track was 0.4 m, and allowed for the extended aerial phases that are characteristic of jerboa locomotion, which we observed. The track was topped with a mesh lid to prevent escape, but the jerboas never came near the vertical limit of the track during forward locomotion.

We revised the methods to address the reviewer’s concern: “The size of the trackway had no detectable effect on the range of speeds used by jerboas when compared to a previous study of unrestricted jerboa locomotion indoors (0.5 to 3.21 m/s) (Schroepfer 1985).”

Also, in contrast to the assertion made in the abstract, it is not clear that this study demonstrates a link between gait transitions and unpredictability of locomotion.

We argue that the frequent transitions between gaits having distinct energetic and biomechanical locomotor functions increases the opportunity for unpredictable locomotion. **To address the reviewer's concerns, we have revised the abstract to more accurately summarize the experimental results:** "Unlike the speed-regulated gait use of cursorial animals to enhance locomotor economy, bipedal jerboa (family Dipodidae) gait transitions likely enhance maneuverability."

I found the methods explaining the entrophy calculation to be incomprehensible. The supplementary methods were equally obtuse. In the context of this study, what are "moments". What empirical data was used and how was it used to calculate entrophy. This section needs to be written in a way that even readers who may not be able to follow the math will be able to gain a sense of what was done.

We appreciate the reviewer's concern and accordingly have **substantially revised the methods and the supplementary methods** in light of the reviewer's comments.

Lines 101 – 114. The Authors state that they examined whether transitions between gaits are associated with speed ranges. But the range of speeds examined (Fig. 1) is less than half the maximum speed of jerboas. Without an analysis of close to the full range of speed, it is not possible to determine if particular gaits are associated with different ranges of speed.

We thank the reviewer for bringing to our attention the imprecise wording of our null hypothesis. **We revised this line to read:** "We first examined whether each jerboa gait is used exclusively at the speed range expected for cursorial locomotion."

Also, Happold (1967) anecdotally reports speeds up to 6.25 m/s (14 mph) in the field, however no citation or description of how speed was measured is provided. We therefore are hesitant to trust this very high speed estimate. To our knowledge, there are no studies that provide reliable measurements of jerboa speed moving in unrestricted natural habitats.

While the data in Fig. 1 may well not encompass the complete speed range of jerboas, we would argue that they are sufficiently broad to refute the null hypothesis provided by the Froude equation: hopping should occur predominantly below 0.54 m/s, skipping should occur predominantly from 0.54 m/s to 1.21 m/s, and running should occur predominantly at speeds above 1.21 m/s. Firstly, all of the locomotion observed by the jerboas was above 0.54m/s. Secondly, the fact that all three gaits are frequently used at sub-maximal speeds is also evidence in support of refuting the null hypothesis provided by the Froude equation. **We revised the results to more clearly state how our findings refute the null hypothesis.**

...Although the maximum speed exhibited by jerboas in the laboratory is lower than the maximum speeds jerboas may exhibit in the field (Happold, 1967), the absence of locomotion at the lowest speed range predicted by the Froude equation, the substantial

overlap between the speed ranges of each gait, and the fact that all three gaits were observed at submaximal speeds, contradict the expectation based on cursorial locomotion that speed regulates gait usage in jerboas.

Lines 162 – 164. The Authors state that they were surprised to find that *D. sagitta* was not significantly more erratic in its locomotion than the jird. The Authors should consider the extent to which the human “predator” actually stressed the jerboas. Compared to the predators actually face, humans are remarkably slow and lumbering. Consequently, the jerboa data may not represent maximum performance.

We agree with the reviewer that humans are rather slow lumbering predators when compared to the normal predators of these rodents. Due to this difference in response time, we used pairs of humans to stress the rodents in the field enclosures. Each human would do their best to “attack” the rodent in the direction of the other human, so that the rodents would rapidly encounter each “predator.” Jerboas and jirds would often jump vertically out of surprise, then perform a rapid bout of locomotion to evade a “predator.” **We have revised the methods section to include a description of this procedure:**

...In light of humans being slower and less maneuverable than the rodents' natural predators, each trial consisted of one human stationed at each end of the enclosure. This had the effect of scaring the rodent towards the other human, increasing the rapidity of each encounter....

There are no published accounts of jerboas interacting with natural predators, so there is no way for us to compare the stress we induced in the jerboas to the stress a normal predator would induce.

While it is possible that our stimulus did not maximally stress the jerboas, we provided a consistent stimulus across all rodent species. Therefore, any difference in behavior must have arisen from the way each species interpreted the stimulus.

Lines 173 – 174. Please present the data discussed in this paragraph in tables or figures.

These data are presented in the spherical plots in Figure 4 B. The colored areas away from the y-axis indicate the likelihood of adjusting the heading of motion in a given direction with respect to the previous location of the animal (e.g. turning). The larger extent in the colored areas in the longitudinal direction (away from the equator) indicates a greater likelihood for vertical motion. In particular, notice that the red clouds on the sphere are much larger in longitude than the green and blue clouds. **We revised the results to improve the clarity of this description:**

...The distributions of angles of motion reveal spatial preferences for each species. Despite more frequent contact with the ground, and therefore more opportunities to change direction in the horizontal plane, quadrupedal jirds showed the greatest

preference for forward locomotion with few turns (Fig. 4 B). In contrast, both jerboa species showed an increased preference for vertical motion (indicated by the longitudinal extent of the colored clouds in Fig. 4 B), suggesting that the evolution of bipedalism likely facilitated the divergence in trajectory unpredictability between jerboas and jirds by allowing jerboas to utilize three-dimensional space and increasing their capacity for acceleration. Differences in entropy between the two jerboa species resulted from the stronger preference of *A. elater* for turning (indicated by the extent of the colored clouds away from the y-axis in Fig. 4 B)...

Assessment of field trajectory unpredictability, lines 299 – 314. How many cameras were used to document the 3D movements? At what framing speed were the videos recorded? How large was the calibrated volume in which analysis was done?

We thank the reviewer for pointing out this oversight. **We revised the methods section to include these details:**

...We illuminated the enclosure using infrared floodlights and recorded animal movements using cameras from which the infrared filter was removed (Casio ZR100). Each animal was filmed in HD (1920 x 1080 pixels) at 30 fps by two cameras as it responded to stimuli (e.g. looming, loud noises) for fifteen minutes or until it presented signs of fatigue...

...The center of the rodent body was tracked in three-dimensional space using DLTdv5 tracking software (Hedrick 2008). The calibrated volume was 4.3 x 2.5 x 0.4 m...

In Supplemental Figures 2 and 3, what are the units of force?

We thank the reviewer for pointing out this oversight. **Units of force (N) have been added to the supplemental figures.**

Supplemental Figure 4 should probably be included in the main body of the manuscript.

Since Figure 4 showed the sampling distributions of simulated data, rather than real data, we had originally placed it in the supplemental information to limit confusion. **Supplemental Figure 4 has since been replaced with a methodological flowchart** that describes the entropy significance test and the sensitivity to tracking error analyses, and includes the sampling distributions for all simulated data.

In this version, **we moved methodological descriptions of experiments that were necessary to obtain the results to the main body of the text** (previously in “Detailed Light-Dark Box Exploration Test Results” subsection). **Supplemental analyses**, such as the The “Entropy Significance Test” and the new “Entropy Divergence Sensitivity to Tracking Error” subsections, **were placed into the supplemental information**, since these involve simulated

data and are not necessary to obtain the results we present in the main body of the text. Now, both the methodological descriptions and the figures describing supplemental analyses are together in the supplemental information.

Reviewer #2 (Remarks to the Author):

This is an interesting study that explores the escape behaviour of bipedal and quadrupedal mammals by developing a novel tool for quantifying the unpredictability of prey locomotor behaviour. The authors are correct in suggesting that this tool offers an opportunity for biomechanists to directly contribute to the field of ecology by linking movement behaviours to predator prey interactions and community dynamics.

The most important weakness of this manuscript is that this tool has not been tested in the field or the lab. The fundamental premise of the paper is that by quantifying the unpredictability of movement by animals then you will be able to provide an alternate metric of escape ability. This may be true but it does need to be tested – merely providing the logic for the metric and developing the math has only provided a first step. My feeling is that the authors should provide a rigorous test of the importance of locomotor unpredictability first before publishing, or the authors should publish this tool in a more specialized journal.

We appreciate the thoughtfulness with which the reviewer considered our submission. However, we would point out that the entropy metric was in fact tested on field data comparing jerboas and jirds (lines 159-162, and Methods subsection “Field Trajectory Unpredictability”). In lieu of testing the unpredictability metric on natural predation events (which would be practically infeasible in this system, as described below), we found that the unpredictability of jerboa and jird escape trajectories in the field varies together with anxiety in open and brightly lit areas. We hope the reviewer finds open-field anxiety to be an appropriate estimate of predator evasion behavior, since rodent anxiety in open areas results from the risk of predation conflicting with the desire to explore or forage (Belzung and Griebel 2001, Lima and Dill 1990, Bailey 2009).

For a more in-depth explanation: Lima and Dill (1990) define predation risk in equation [1]: $\text{Probability}(\text{death}) = 1 - \exp(-\alpha \cdot d \cdot T)$. (α) is the rate of encounter between predator and prey, which depends on the openness of the environment. (d) is the probability of death *given* an encounter, which depends on their escape ability (equation [2]). (T) is the time in that area. Therefore, if a prey animal finds itself in an open area where the probability of encountering a predator (α) is high, and the prey animal has a low predator evasion ability (high d), the prey's only option to minimize their $\text{Probability}(\text{death})$ is to minimize their time (T) in the open area. On the other hand, a different prey animal with greater predator evasion ability (low d), with the same predator and in the same open area (α), can stay in the area for a longer amount of time (T), without significantly increasing their $\text{Probability}(\text{death})$ with respect to the less agile prey. According to this equation, for similar species that encounter the same predators, time spent in an open area must vary inversely to the predator evasion ability of the prey to maintain a certain level of safety.

Unfortunately, obtaining precise and quantitative data on predator-prey interactions involving jerboas is currently practically infeasible. Jerboas are extremely rare in North American laboratories, so terminal experiments would be impractical. Furthermore, an experimental protocol including a staged predator-prey encounter on live animals (as in Longland and Price, 1991) is not likely to be approved by our IACUC, given current ethical standards. In the field, it is nearly impossible to predict when a jerboa will encounter a predator. This is because jerboa encounters in the field are unpredictable and rare, as jerboas become startled and rapidly flee when encountered. Happold (1967) states "it was not possible to study activity in wild-living jerboas since they are nocturnal and widely scattered in the desert." In fact, there are no published descriptions of a natural jerboa predation event in the wild. Even if a few chance encounters were to be recorded, it is highly unlikely that these data would yield sufficient precision or statistical power to provide a reliable analysis of locomotor behavior under these conditions.

Because our only option was to measure predator evasion ability indirectly, **we have revised the abstract, introduction, and results, to clarify the link between predator evasion ability and open field anxiety.**

(Abstract) ...Consistent with this hypothesis, jerboas exhibit less anxiety in open fields than quadrupedal rodents, a behavior that varies inversely with predator evasion ability.

(Introduction) ...Because exposed microhabitats are an important source of nutrient resources, there is a conflict in small foraging animals between exploration and risk of predation that determines how long an animal will stay in an open area (Lima 1990). Enhanced evasion ability decreases the risk of predation in exposed microhabitats, resulting in an inverse relationship between predator evasion ability and thigmotaxis --- the behavioral affinity to shelter (Kotler 1984a, Hendrie 1998, Lima 1990). Thigmotaxis can therefore be used to indicate relative evasion ability between similar animals that encounter the same predators...

(Results) Small, foraging animals that are susceptible to predation are subject to a conflict between the desire to explore new areas to discover food and the desire to remain in covered areas safe from predators. Due to this conflict, the amount of time an animal spends in an open area varies with their predator evasion ability (see Equation (1) in (Lima 1990)), assuming that there is a minimum acceptable probability of death. Therefore, to determine whether prey with unpredictable trajectories have greater predator evasion ability, we measured rodent thigmotaxis...These results corroborate the pattern we found in the natural habitat and suggest bipedalism is associated with less thigmotaxis and greater predator evasion ability in open microhabitats.

A potential shortcoming of the study is the limited number of species that have been observed. Comparing two bipedal species with just a single quadruped does not really provide us with the

confidence that all bipedal species have more unpredictable locomotion than equivalent sized quadrupeds. This is especially important when one considers that there are such substantial differences between the two bipedal species that were studied.

It is true that there were substantial differences in entropy and directionality between the two jerboa species, but importantly, the entropy of both jerboas was greater than the entropy of the quadrupedal jird. If we assume that all desert rodents must move in the horizontal plane (to forage, find mates, etc), then any movement in the vertical direction results in an increase in the entropy of the rodent trajectory.

Although jerboas are morphologically more derived than N. American kangaroo rats and Australian hopping mice (Moore et al. 2015), several studies regarding the biomechanics of bipedal desert rodents suggest that significant movement in the vertical plane is important for predator evasion in these other bipedal rodents.

Longland and Price (1991) performed an experiment in which they directly observed the predation of bipedal and quadrupedal American rodents by Great Horned Owls. They stated that although bipedal rodents were struck more frequently because they were more likely to be in open areas, they used quick maneuvers and vertical jumping to evade predation. These bipedal rodents evaded predation with greater success than quadrupedal rodents in the same enclosure. This study supports our hypothesis that bipedality enhances predator evasion with respect to sympatric quadrupedal rodents. Furthermore, Biewener and Blickhan (1988) showed that kangaroo rat hindlimb tendons are specialized for maneuverability and rapid acceleration, and kangaroo rats reliably performed rapid vertical jumping for this experiment. Similarly, Webster (1962) was able to induce rapid vertical jumping by simply presenting kangaroo rats with the auditory cues of a their predators. These biomechanical and behavioral studies demonstrate that kangaroo rats are specialized for jumping in the vertical plane when stressed. While Australian hopping mice are less thoroughly studied, it is known that they are facultatively bipedal, and only hop bipedally when chased at their highest speeds (Marlow 1969). This shift from quadrupedal to bipedal locomotion during stressful circumstances increases the dimensionality of their motion. We believe that these studies provide strong evidence that all groups of bipedal rodents exhibit movement in the vertical direction during evasive maneuvers.

We have revised the introduction and discussion of the paper to state more clearly why our findings in jerboas may be applicable to other systems of rodent bipedalism, and have added references to support this claim.

(Introduction) ...While this study is limited to one example of bipedalism in rodents, kangaroo rats and Australian hopping mice have similar biomechanical and ecological divergence from sympatric quadrupeds that may also be explained by the divergence in trajectory unpredictability we measured in jerboas (Biewener1988, Clark2016, Baudinette1976, Dickman2010)....

(Discussion)...Ricochetral bipedal locomotion in rodents may be an adaptation to enhance predator evasion (Bartholomew1951a, Schropfer1985). For example, Australian hopping mice transition from quadrupedal to bipedal locomotion during evasive maneuvers (Marlow1969). Kangaroo rats are obligately bipedal and produce evasive maneuvers in response to predator cues, and have greater evasion success than sympatric quadrupedal rodents (Clark2012, Webster1962, Longland1991). To determine whether the shift to bipedal locomotion enhances predator evasion, we developed metrics for quantifying the evasiveness of a maneuver. Since bipedal locomotion with large aerial phases increases the dimensionality of rodent locomotion, we sought out analytical methods capable of quantifying the contributions of both velocity and dimensionality to identify differences observed between bipedal and quadrupedal rodent locomotion.

...The substantial differences we have recorded in trajectory unpredictability and open-field anxiety between sympatric jerboas and jirds (Figure 5), along with several studies noting similar biomechanical and ecological divergence between bipedal and quadrupedal rodents on other continents (Biewener 1988, Clark 2016, Baudinette 1976, Dickman 2010)...

One aspect of your manuscript that got me thinking is the relationship between the “increased” unpredictability of locomotion for the two bipeds and its possible link with their probability of being captured by predators. Your metric of entropy seems to be quantifying (I’m not absolutely sure of this, so this could be further explained) the probability that the animals will be in a certain location in the next time point given their current position. Clearly, the two bipeds can have greater changes in direction when in contact with the ground but they are also more likely to spend a greater amount of time in the air (flight time) when they absolutely cannot change directions. In contrast, the quadrupeds may be more predictable in their location at each time point, but they have more opportunities to change directions at any point in time. The quadrupeds may only be able to make small changes in direction when in contact with the ground, but they spend more time on the ground, thus allowing more opportunities for changes in direction. I’m then left wondering what these different locomotor attributes for the different species have for their probability of being captured by predators. This point reinforces to me the importance of testing the ‘benefits’ of your metric of locomotor unpredictability for escape success. I suspect this metric of entropy will be just one of many affecting escape success. Having said that, I don’t think this detracts from the novelty of your tool for studies of escape success but it does limit the novelty of this study when it is not tested empirically.

We appreciate the reviewer’s insightful analysis of the kinematic differences between bipedal and quadrupedal rodent locomotion, raising a key point that *horizontal* movement direction cannot change when animals are airborne versus on the ground (in the vertical direction movement direction does change but of course also follows a predictable parabolic trajectory). Because our data are sampled at regular intervals in time (the reviewer’s interpretation of our analysis is correct, which we have attempted to make clear in our revised MS), our analysis includes **both** when the animal is in the air and in contact with the ground, and therefore directly

addresses this issue. Although quadrupeds may have more opportunities to change direction in the horizontal plane over a given amount of time due to their more regular contact with the ground, our data (Figure 4b) show that quadrupedal rodents do not take advantage of this, and rather prefer to move in straight lines in the horizontal plane. **We revised the results to highlight this important point.**

Despite more frequent contact with the ground, and therefore more opportunities to change direction in the horizontal plane, quadrupedal jirds showed the greatest preference for forward locomotion with few turns (Fig. 4 B)

Jerboas and all jumping animals change *vertical* movement direction at the apex of each aerial phase by following a parabolic ascending to descending trajectory (this is why the spheres of data in Figure 4B are symmetric about the equator). This preference for vertical motion is one that is not as exploited by the quadrupedal rodent. Our analysis of movement unpredictability in three-dimensional space highlights this difference. Previous papers comparing maneuverability between bipedal and quadrupedal rodents simply examined their footprints, thus ignoring the additional dimension available to the bipedal rodents (Djawdan and Garland 1988, Djawdan 1993). We argue that by evolving bipedality, this additional vertical dimension of movement increases the possible locations available to rodents. Importantly, we present evidence to show that different species of bipedal rodents diverge in their unpredictability by the combination of diverging in the frequency with which they turn in the horizontal plane and the frequency with which they move in the vertical direction. **We revised the results to accentuate this point.** We are hopeful that our revisions and responses satisfactorily address the reviewer's concern regarding these important points.

...In contrast, both jerboa species showed an increased preference for vertical motion (indicated by the longitudinal extent of the colored clouds in Fig. 4 B), suggesting that the evolution of bipedalism likely facilitated the divergence in trajectory unpredictability between jerboas and jirds by allowing jerboas to utilize three-dimensional space and increasing their capacity for acceleration. Differences in entropy between the two jerboa species resulted primarily from the stronger preference of *A. elater* for turning (indicated by the extent of the colored clouds away from the y-axis in Fig. 4 B) and to a lesser extent a difference in the propensity for vertical motion...

Reviewer #3 (Remarks to the Author):

The authors present a novel approach to assess behavior in the lab and in the field, and support conclusions regarding the evolution of bipedal hopping in desert rodents. In general, the paper is very well written and the rationale for the approach is clear. This work is both highly interesting and potentially very important. However, I do feel the authors overstate their conclusions somewhat in regards to the evolution of bipedal hopping. Specifically, as stated by the authors, jerboas are the most derived of the bipedal desert rodents. Therefore, it's not clear

how representative their behavior is of other species, or how well it explains the evolution of bipedality.

We hope that our response to Reviewer 2 above, who raised the same concern, has addressed this point.

While I do believe the approach taken by the authors provides interesting and novel insight into the behavior of jerboas, I would caution against overstating the role of unpredictability in the evolution of bipedal locomotion. For example, it is still likely that factors such as acceleration capacity also contributed to the evolution of bipedal hopping and may have in fact preceded the capacity for unpredictability.

We agree there are likely several metrics of locomotor performance that may be relevant to predator evasion. However, many of these metrics are measures of biomechanical capacity, or *potential* limits to locomotor performance. In this paper we argue that *observed* trajectory unpredictability is a valuable biomechanical metric to quantify with respect to predator evasion and evolution. This is because the morphological changes in functional capacity determine the limits of animal motion, but the unpredictability of a trajectory is a behavioral utilization of the morphological function that actually determines predator evasion, and is therefore more relevant to evolutionary fitness. In other words, if changes in accelerative capacity (or other performance metrics) are contributing to predator evasion ability, then this will be observed and will inform the entropy calculation. **We have revised the results to clarify this distinction:**

While morphology determines the *capacity* to generate complex behaviors (i.e. maneuverability), this only defines the limits of theoretical performance (Webb 1983, Norberg 1994). Predator-prey interactions are determined by the *observed* performance of locomotor trajectories that result from path planning behavior, regardless of the animal's theoretical maneuverability (Moore 2015).

I also have some minor points for clarification. Please see below.

In equation 2, how is leg length determined? This can vary greatly in species with a crouched posture.

Our definition in line 103 of the original text (mean hip height at mid stance) may have been vague. **We revised the methods to describe this more thorough estimate of leg length.**

Leg length was estimated as the height of the hip at mid-stance for each trial. For n=80 trials, the mean leg length was 0.0605 m, 0.0066 m standard deviation.

You might consider using the square root of Fr (i.e., $u/(gl)^{0.5}$), as this eliminates the exponential. However, if the goal is to compare to older previously published data, this may not be necessary.

We appreciate the suggestion of the square root of Fr , but as the reviewer also suggested, would prefer to compare our use of Fr to previously published data.

What feature on the animals was tracked in the field trajectory trials? If it was just the animal as a whole, how does position error effect the entropy calculation, i.e., how does digitizing a different position on the animal from frame to frame effect error?

The center of the body was tracked. Digitizing a slightly different position on the animal has a negligible effect on the divergence in entropy between the species because the magnitude of the error is quite small compared to the trajectories of the animal. Furthermore, **we have added a new section in the supplementary methods that provides a sensitivity analysis to determine the impact of tracking error on our results.** This analysis demonstrates that random error increases the entropy of each distribution, as expected, but the relationship between the entropy of each species remained unchanged. **We also replaced the supplemental figure** showing the p-value sampling distributions with a flow chart describing how the significance test and the sensitivity to tracking error analyses were performed.

Reviewer #3 (Remarks to the Author):

In general, I believe that the authors have addressed my major comments. I am still not entirely convinced that the comparisons made using the derived species generalize to the evolution of hopping across species. However I will leave it to readers to make up their own minds.

Reviewer #4 (Remarks to the Author):

My reading of this paper is that it differ from previous studies in that measuring manoeuvrability as entropy includes both biomechanical limits, and behavioural limits. This makes it interesting and novel from previous studies. I also think that this technique is as ecologically relevant as any other method for determining manoeuvrability. This technique may become more widespread as it is more simply measured than other methods which require an animal to repeat a performance measure many times. This alone should make it a valuable contribution to science.

The paper appears to have been thoroughly reviewed by the previous three reviewers and I don't have any further suggestions. The authors responses to the reviewers seem appropriate to me, though I hope they have made attempts to make their methods as understandable as possible.

Reviewer #3 (Remarks to the Author):

In general, I believe that the authors have addressed my major comments. I am still not entirely convinced that the comparisons made using the derived species generalize to the evolution of hopping across species. However I will leave it to readers to make up their own minds.

We thank the reviewer for their additional examination of our manuscript. Our objective is to provide a mechanistic explanation in support of the theory that bipedality evolved in desert rodents to enhance predator evasion and limit interspecific competition that was previously presented by Rogovin (1999), Kotler (1994), and Schröpfer and Klenner-Fringes (1991).

Reviewer #4 (Remarks to the Author):

My reading of this paper is that it differ from previous studies in that measuring manoeuvrability as entropy includes both biomechanical limits, and behavioural limits. This makes it interesting and novel from previous studies. I also think that this technique is as ecologically relevant as any other method for determining manoeuvrability. This technique may become more widespread as it is more simply measured than other methods which require an animal to repeat a performance measure many times. This alone should make it a valuable contribution to science.

The paper appears to have been thoroughly reviewed by the previous three reviewers and I don't have any further suggestions. The authors responses to the reviewers seem appropriate to me, though I hope they have made attempts to make their methods as understandable as possible.

We thank the reviewer for examining our manuscript in the context of the predator-prey biomechanics.

Editor's summary:

Biomechanical understanding of animal gait and maneuverability has primarily been limited to species with more predictable, steady-state movement patterns. Here, the authors develop a method to quantify movement predictability, and apply the method to study escape-related movement in several species of desert rodents.

This looks good.

Abstract:

Mechanistically linking movement behaviors and ecology is key to understanding the adaptive evolution of locomotion. Predator evasion, a behavior that enhances fitness, may depend upon short bursts or complex patterns of locomotion. However, such movements are poorly characterized by existing biomechanical metrics. We present methods to quantitatively characterize the unpredictability of non-steady-state locomotion. We then apply the method by examining sympatric rodent species whose escape trajectories differ in dimensionality. Unlike the speed-regulated gait use of cursorial animals to enhance locomotor economy, bipedal jerboa (family Dipodidae) gait transitions likely enhance maneuverability. In field-based observations, jerboa trajectories are significantly less predictable than those of quadrupedal rodents, likely increasing predator evasion ability. Consistent with this hypothesis, jerboas exhibit less anxiety in open fields than quadrupedal rodents, a behavior that varies inversely

with predator evasion ability. Our unpredictability metric expands the scope of quantitative biomechanical studies to include non-steady-state locomotion in a variety of evolutionary and ecologically significant contexts.

We prefer to retain “quantitatively characterize” instead of “quantify” because this method not only measures entropy, but also describes how this entropy value is achieved, thus characterizing the locomotor behaviors.

We agree with the reviewer’s suggestion to break up the fourth sentence. We have edited the suggested sentence to state that the sympatric rodent species use escape trajectories that differ in dimensionality because the sympatric rodent species’ escape trajectories are not variable in the same way.

We agree with all other changes to the abstract.

Overall Comments:

Make sure that all figures and tables, main and supplementary, are referred to at least once in the manuscript. Also, please make sure that the first referral to any given figure is in numerical order (i.e. you should not be referring to fig 2 before you mention Fig 1, or to supplementary Fig 2 before Supplementary Fig 1)

Added a reference to Supplementary Table 1

Please adjust all figure callouts to use the formatting: Fig. 1, Figs 1-2

Please modify the names of Supplementary content as follows: Supplementary Fig. 1, Supplementary Table 1

Figure callouts have been adjusted

Please change all citations to superscript to avoid mistakes during typesetting.

All citations have been changed to superscript

Results:

Please provide additional relevant test statistics (F value, degrees of freedom) for this ANOVA and those that follow.

Anova F value and df have been included

We don't allow the use of italics for emphasis. Please modify here and elsewhere, for example by using single quotation marks instead of italics.

Italics changed to single quotation marks

(Line 175) Where results of statistical tests are stated (eg p values), please also state the name of the statistical test and other relevant test statistics.

The statistical test used to generate p-values for the entropy measurements is a modified t-test, with the methods thoroughly described in the Supplementary Methods and in Supplementary Fig. 3

(Line 220) The test and relevant statistics are now stated.

Methods:

Please add a subheading between Methods and the first paragraph.

Paragraph subheading in Methods was added

Please provide sufficient detail for the method to be repeated, even it has been published previously.

The procedure is described in the next paragraph. The anxiety test reference has been moved to immediately precede this description.

Acknowledgements:

It is journal policy not to include referees in the acknowledgements.

Removed

Figures:

(Table 1) I find this Table a bit confusing. A few suggestions:

1- Under each of the primary column headings, I recommend you add sub-headings for p value and F statistic so it is clear to readers without needing to refer to the legend. It also seems more conventional to list F then p values, reading left to right. Finally, it would also be prudent to provide degrees of freedom somewhere.

F and P values have been swapped, as requested, and labeled in the table. df was added to the caption.

2- Our typesetters may get confused by where to place the headings (vertical versus fore-aft forces) relative to the lines. They will probably need to make them left-justified headings to comply with formatting standards.

The titles for each sub-table were adjusted to aid the typesetters.

(Figure 1) Please add an x axis label (e.g. Gait type).

“Gait” was added to the x-axis

(Figure 2) Should there be correlation strength and/or P values to report for this path analysis? Can you mention in the legend what analysis the probability values are obtained from?

Changed “probability” to “frequency” as no statistical tests were performed. The path thickness represents the number of times one gait led to another gait in 36 trials.

(Figure 4) I would recommend parsing out panel B into three separate labeled panels so that, for each one, you can provide a note in the legend about which species it is (perhaps using the common names which are used more in the text than the latin names), and whether it's quadupedal or bipedal.

Just a suggestion.

Each sphere is now a separate subfigure. The pedality and the common name for each species has been added to the figure caption.

More importantly, please, somewhere in the legend text describe the color scheme. I also suggest altering the color scheme somewhat because the lines for *D. sagitta* and *A. elater* in panel A might be indistinguishable to colorblind readers.

Figures 4 and 5: New colorblind-friendly colors have been selected with the aid of colorbrewer2.org

(Figure 5) Please describe which color denotes which species in this legend, rather than referring to a different legend.

Color description has been added to the legend.

Supplementary Information:

Please note that you **will not** have an opportunity to revise the SI further once the manuscript is accepted.

Line numbers have been removed.

Please note that the SI will need reformatting regarding the following issues:

1. We don't include header information (date, title, etc).

Header has been removed

2. Rather than numbered sets of headings and subheadings, please use only the following named content types (preferably in the order listed):

Supplementary Figures, Supplementary Tables,
Supplementary Notes, Supplementary Methods,
Supplementary Discussion,
Supplementary References

Numbered sections have been removed, and sections have been rearranged as requested.

*Once these are all updated, please be sure to update call-outs in the main text and in this document.

These have been updated

3. The lists of formulas and numbered references should start at 1 and be self-contained.

References and equations begin with 1

As in the main text, we prefer that there are not more than 2 'tier's of subheadings'.

Third tier of section hierarchy has been removed

□

The square notation is placed at the end of a mathematical proof to indicate its completion. See https://en.wikipedia.org/wiki/Q.E.D.#Symbolic_forms_in_typography

Please rename as Supplementary Notes or Supplementary Discussion.

Supplemental Results has been renamed Supplementary Notes.

References has been renamed Supplementary References

(Supplementary Algorithm 1) I recommend renaming this bit of text as a Supplementary Note, and replacing the reference to it in the main text accordingly.

Supplementary Algorithm 1 has been placed within Supplementary Methods.

(Supplementary Figure 1 and 2) Please name the statistical test in the legend, since P values are given. (This applies to all other legends as well).

Supplementary Figure 1 and 2 have "Gait" added to the x-axis. F, degrees of freedom, and statistical test have been added to the figure legends.

The colors of Supplementary Figure 3 have been changed to match Figure 4 and 5.

(Supplementary Table 1) Please name the statistical test.

The statistical test has been added.